# Reproductive Stage Drought Tolerance in Wheat: Importance of Stomatal Conductance and Plant Growth Regulators

**DOI:** 10.3390/genes12111742

**Published:** 2021-10-29

**Authors:** Olive Onyemaobi, Harriet Sangma, Gagan Garg, Xiaomei Wallace, Sue Kleven, Pipob Suwanchaikasem, Ute Roessner, Rudy Dolferus

**Affiliations:** 1CSIRO Agriculture and Food, GPO Box 1700, Canberra, ACT 2601, Australia; xiaomei.wallace@csiro.au (X.W.); susan.kleven@csiro.au (S.K.); rdolferus58@gmail.com (R.D.); 2CSIRO Centre for Environment and Life Sciences, 147 Underwood Avenue, Floreat, WA 6014, Australia; gagan.garg@csiro.au; 3Grains Research and Development Corporation, National Circuit, Barton, ACT 2600, Australia; Harriet.Sangma@grdc.com.au; 4School of BioSciences, The University of Melbourne, Melbourne, VIC 3010, Australia; psuwanchaika@student.unimelb.edu.au (P.S.); u.roessner@unimelb.edu.au (U.R.)

**Keywords:** drought stress, wheat, stomatal conductance, spike development, transcriptome, ABA, auxin, cytokinin, isohydric, anisohydric

## Abstract

Drought stress requires plants to adjust their water balance to maintain tissue water levels. Isohydric plants (‘water-savers’) typically achieve this through stomatal closure, while anisohydric plants (‘water-wasters’) use osmotic adjustment and maintain stomatal conductance. Isohydry or anisohydry allows plant species to adapt to different environments. In this paper we show that both mechanisms occur in bread wheat (*Triticum aestivum* L.). Wheat lines with reproductive drought-tolerance delay stomatal closure and are temporarily anisohydric, before closing stomata and become isohydric at higher threshold levels of drought stress. Drought-sensitive wheat is isohydric from the start of the drought treatment. The capacity of the drought-tolerant line to maintain stomatal conductance correlates with repression of ABA synthesis in spikes and flag leaves. Gene expression profiling revealed major differences in the drought response in spikes and flag leaves of both wheat lines. While the isohydric drought-sensitive line enters a passive growth mode (arrest of photosynthesis, protein translation), the tolerant line mounts a stronger stress defence response (ROS protection, LEA proteins, cuticle synthesis). The drought response of the tolerant line is characterised by a strong response in the spike, displaying enrichment of genes involved in auxin, cytokinin and ethylene metabolism/signalling. While isohydry may offer advantages for longer term drought stress, anisohydry may be more beneficial when drought stress occurs during the critical stages of wheat spike development, ultimately improving grain yield.

## 1. Introduction

Drought stress is a major constraint to crop productivity. Our main food crops, the cereals, suffer dramatic yield losses when drought spells coincide with the reproductive growth phase. Climate change scenarios predict increasing occurrences of irregular rainfall and associated drought spells in the future. Enhancing drought tolerance of cereals and other food crops has become essential to stabilise and secure food supplies for an increasing world population [1,2,3]. 

Breeding for improved yield performance of wheat under water limiting conditions has been a continuous effort in Australia—and globally. While incremental progress has been made over the last decades, major breakthroughs have been lacking and reliable markers for marker-assisted selection are still missing [4,5,6]. Morphological and physiological traits that improve root development and vegetative growth have traditionally been used to enhance drought tolerance in wheat: transpiration efficiency, water use efficiency, stomatal conductance, osmotic adjustment, xylem cavitation resistance [7,8,9,10,11]. These ‘vegetative’ traits do not necessarily improve reproductive stage drought tolerance, nor do they guarantee higher grain yields under terminal drought conditions [12,13,14,15]. Very few drought tolerance traits focus on the reproductive phase, even though protecting reproductive development is considered essential to secure productivity of wheat crops [15]. Phenology of flowering time as a stress escape mechanism has played an important role in protecting crops against seasonal droughts [5,16,17,18,19]. A multitude of candidate genes have been identified to improve drought-tolerance via transgenic approaches, but only few transgenes that improved drought tolerance in controlled environment assays convincingly contributed to drought tolerance in field trials [20,21,22,23]. In wheat, improvements in management and growing practices, such as controlling sowing time and phenology, may arguably have contributed more to improving cereal productivity under water-limited conditions than genetic gain [24,25]. 

Both vegetative and reproductive growth are continuously adjusted to daily, seasonal or temporal depletion of water availability and other environmental factors [26,27,28]. Plant growth depends on the homeostasis and hydraulic effect of water and the flow of water and nutrients in the vascular system. The regulation of transpiration by stomata is essential to regulate photosynthetic activity, as well as transport of photosynthates throughout the plant [29,30,31,32,33]. Traditionally, plants have been classified in two groups depending on their hydraulic behaviour [34,35,36]. Isohydric (water-saving) plants maintain water potential by reducing stomatal conductance and transpiration. Anisohydric (water-wasting) plants osmotically adjust to maintain cell turgor by increasing cellular potassium levels and by synthesising osmolytes such as sugars and amino acids [37,38]. The riskier anisohydric approach sustains growth and development under shorter-term droughts, while the more conservative isohydric response is better for passive survival under longer term droughts [34,39,40]. During evolution of land plants, acquisition of stomata in bryophytes and lycophytes and recruitment of the phytohormone abscisic acid (ABA) for regulation of stomatal conductance are both linked [41,42]. Stomatal closure restricts gas exchange, ultimately leading to repression of photosynthetic activity; it also restricts the transpiration stream and transport of photosynthates [9,43,44]. In reproductive tissues, ABA represses sugar supply and sink strength in anthers, causing abortion of pollen development and sterility under drought conditions [45]. The negative effects of ABA on reproductive development contradict the perceived beneficial effects of the hormone on drought tolerance at the vegetative stage [43,46,47]. The role of ABA in the response to drought remains therefore controversial and in need of further research [48].

The aim of this paper was to investigate in more detail the drought response of two wheat lines we have previously shown to differ in drought-tolerance at the reproductive stage [49]. An important observation was that the two wheat lines diverge in terms of drought-induced ABA accumulation, as ABA was observed to increase in reproductive tissues of the drought-sensitive line but not in the drought-tolerant line. The ABA accumulation in the sensitive line caused reduction in anther sink strength and pollen sterility [45,49,50,51]. In this study we show that the difference in drought-induced ABA response in the spike of the two wheat lines also extends to differences in regulation of stomatal conductance in the vegetative plant parts. The drought-sensitive wheat line induces ABA synthesis and shows isohydric behaviour, while the tolerant line is initially anisohydric but becomes isohydric under more severe drought stress conditions. Gene expression profiling studies using spikes and flag leaves of both wheat lines revealed significant differences in gene expression between the two wheat lines and the two tissue types. The results indicate that interactions between four main growth-regulating hormones (ABA, auxin, cytokinin and ethylene) may be responsible for the observed differences in drought tolerance. 

## 2. Materials and Methods

### 2.1. Controlled Environment Wheat Growing Conditions and Drought Treatments

The wheat cultivars used in this study, drought-tolerant Halberd and drought-sensitive Cranbrook, were previously described [49]. Screening of a Cranbrook x Halberd doubled haploid (DH) population for QTL mapping previously identified the DH lines CH67 and CH115 as the most tolerant and sensitive lines respectively of the mapping population [52,53]. The two Australian commercial lines, Westonia and Excalibur, were shown to be drought-tolerant using our standard drought-tolerance assay (Dolferus et al., unpublished data). The procedure for growing and drought-stressing the wheat plants was as described previously [49,54]. Briefly, twelve seeds were germinated in rectangular trays filled with composted soil and plants were drought-stressed at the young microspore (YM) stage of pollen development. Tillers with YM stage spikes were tagged and plants were stressed for two days in a controlled environment cabinet using the water withholding method. The trays were then re-watered and spike sterility was scored at maturity. For the experiments described in this paper, the previously described method was modified to improve reproducibility and severity of drought conditions. The soil in the trays was dried out for one week in a ventilated growth chamber and the dry soil weight in each tray was adjusted to the same weight (11 Kg). Before sowing, the trays were rehydrated to the same total weight (12 Kg). Plants were grown in a Conviron PGC20 controlled environment growth chamber using the following temperature and lighting profile: 24 °C/16 °C (light/dark), 16 h/8 h light/dark cycle and 400 μmol.m^−2^.s^−1^ light intensity. The trays were watered once daily with 500 mL of water and the weight was adjusted once weekly to maintain the same degree of soil water saturation between the trays. When ~10 tillers per tray reached the YM stage, YM-stage tillers were tagged and watering was stopped for two days or longer—as indicated in the Results section. The YM stage of pollen development was determined using auricle distance (AD) measurements as described previously [49,54]. Drought-treated trays were re-watered and spike grain numbers and spikelet numbers were determined at maturity.

### 2.2. Measurements of Stomatal Conductance and Stomatal Density

Stomatal conductance measurements were carried out on the flag leaves of tillers that reached the YM stage, using a SC-1 leaf porometer (Decagon Devices, Pullman, WA, USA). The porometer was calibrated according to the instruction manual and readings were collected from the adaxial and abaxial side of at least five different flag leaves at each time point. The data were exported to Excel for analysis. Stomata from wheat flag leaf material were observed using a Zeiss EVO LS15 scanning electron microscope (SEM). Fiji software [55] was used to determine stomatal density from SEM pictures taken at 62x magnification. The Fiji line tool was used to measure the length of stomatal guard cells.

### 2.3. RNA Extraction, Gene Expression and RNA Sequencing Procedure

Three biological repeat samples were harvested from YM-stage spikes and flag leaves of both control and drought-stressed Cranbrook and Halberd plants. Plant RNA for real-time RT-PCR gene expression studies was isolated using a Qiagen RNeasy Plant kit according to manufacturer’s instructions (Qiagen, Chadstone, Australia). Genomic DNA traces were removed using RQ1 DNase treatment (Promega, Madison, WI, USA) and RNA was precipitated using 2M LiCl. Thermoscript reverse transcriptase (Invitrogen, Carlsbad, CA, USA) was used to reverse-transcribe 0.5μg aliquots of total RNA per sample. Primers for the ABA biosynthetic genes *TaNCED1* and *TaNCED2* encoding 9’-cis-epoxycarotenoid dioxygenase were derived from wheat EST clones CD884104 and CA731387, respectively [45]. BLASTN searches against the wheat genome (TGACv1 and IWGSC) indicated that the genes with highest sequence identity to *TaNCED1* and *TaNCED2* are located on chromosome 5B (TRIAE_CS42_5BS_TGACv1_423804_AA1383550.1; TraesCS5B02G029300.1) and 5A (TRIAE_CS42_5AL_TGACv1_374601_AA1204180.1; TraesCS5A01G374000.1), respectively. We used the wheat β-actin1 gene for normalization of the gene expression results. PCR primers were designed using Geneious R11 bioinformatics software (https://www.geneious.com/, accessed on 20 June 2021). All primer sequences are provided in Appendix A.

For RNA sequencing, poly(A^+^) mRNA was isolated using a Qiagen Oligotex mRNA kit (Qiagen, Chadstone, Australia). RNA sequencing (Illumina bcl2fastq 2.17.1.14, San Diego, CA, US) and preliminary bioinformatics analysis was carried out by the Australian Genome Research Facility (AGRF; www.agrf.org.au, accessed on 20 June 2021) according to ISO17025: 2005 requirements. The data yields for RNA-seq sample are provided in Appendix A. Q30 quality scores of >92% were obtained across all samples. RNA-seq reads were checked for quality and trimmed for Illumina adapters using BioKanga ngsqc (https://github.com/csiro-crop-informatics/biokanga, accessed on 20 June 2021). This was followed by mapping of reads to the Chinese Spring reference genome sequence (v1.0) using BioKanga Align with sensitive settings (parameters: --mode=1 --minchimeric=50 --substitutions=5 --pcrwin=250 --snpreadsmin=10). Mapped reads were allocated to genomic map loci using BioKanga and default settings. Read counts from different biological replicates and samples were combined for each gene and the resulting count matrix was generated using BioKanga gendeseq for input to EdgeR [56]. The count matrix was normalized and analysed for differential expression using the EdgeR package in R version 3.2.2 (http://www.r-project.org/, accessed on 20 June 2021). We used 0.05 as the false discovery rate cut-off to determine differentially expressed genes (DEG). Enrichment for Gene Ontology (GO) terms in the DEG lists was evaluated using TopGO software in R. Significance of enrichment of GO terms was determined with Fisher’s exact tests using both the weight and classic methods and was performed using TopGO v 2.38.1 [57]. To compare DEGs between wheat lines and tissues, Venn diagrams were constructed using an on-line Draw Venn Diagram tool (http://bioinformatics.psb.ugent.be/webtools/Venn/ accessed on 20 June 2021).

Since functional annotation of the wheat genome is still in progress, we used BLASTP peptide sequence similarity searches against the rice and barley peptide databases to assign functional annotations to the wheat DEGs. The rice and barley protein databases were downloaded from Ensembl Plants and searches were carried out using a standalone version of BLAST (v.2.2.30; E-value < 1 × 10^−5^). All the sequences of unique hits were then used to BLAST search against the proteome database of the first available TGACv1 wheat genome annotation [58]. Gene names and general descriptions, along with Gene Ontology (GO) information were extracted from the most significant matches in the database. For the data analysis we only relied on functional annotations that were the same or very similar (e.g., transcription factors, signalling factors) for both barley and rice. The CDS sequences of TGACv1 gene loci were used in BLASTN searches to identify the corresponding IWGSC v.2 gene annotations (See Appendix A). 

### 2.4. Phytohormone Quantification

Phytohormones were measured for YM stage spikes and flag leaves of both control and drought-stressed plants (two-days of drought). Three biological samples were collected from each tissue and analysed three times using a targeted metabolomics approach. A detailed protocol of quantitative analysis using liquid chromatography-mass spectrometry (LC-MS) machine and the list of all plant growth regulators that were analyzed is provided in the ‘Appendix A’ file. Briefly, approximately 100 mg of lyophilized tissues were mixed with 70% methanol supplied with six phytohormone internal standards. The mixtures were shaken using a Precellys 24 tissue homogenizer (Bertin Instruments, France) to extract metabolites from the tissues and then centrifuged at 13,000 rpm for 15 min. Supernatant was transferred to a glass vial and subjected to Triple-Quad 6410 LC-MS machine (Agilent Technologies, Santa Clara, CA, USA). Separation was performed on Eclipse XDB-C18 column (1.8 µm, 2.1 × 100 mm) at 45 °C. Details about the LC gradient program and MS detection method can be found in the ‘Appendix A’ file. 

## 3. Results

### 3.1. Drought Stress Does Not Induce ABA Biosynthesis Genes in Drought-tolerant Halberd Spikes

We previously studied the effect of drought stress at the YM stage of pollen development using drought-tolerant line Halberd and sensitive line Cranbrook [45,49]. In this paper we extend this investigation to both reproductive (YM-stage whole spikes) and vegetative tissues (flag leaves) using the same two wheat lines. At the YM stage, the spike is actively developing while surrounded and supported by the leaf sheath of the flag leaf and penultimate leaf (Figure 1A). We tested whether expression of the *TaNCED1* and *TaNCED2* genes encoding the ABA biosynthesis enzyme nine-cis-epocycarotenoid dioxygenase (NCED) was the same in the whole spike compared to what we previously observed in dissected anthers. *TaNCED1* and *TaNCED2* were both induced 2- to 3-fold by drought stress in the sensitive variety Cranbrook, but there was no significant effect on the expression of these genes in the tolerant variety Halberd (Figure 1B). This expression pattern is the same of what we previously observed for dissected YM stage anthers of the same two wheat lines [45]. 

### 3.2. Drought-Tolerant Wheat Maintains Stomatal Conductance Longer under Drought Conditions

The difference in expression of *NCED* genes in drought-stressed Cranbrook and Halberd prompted us to look for differences in stomatal conductance in flag leaves under drought conditions. The flag leaf is the last and youngest leaf to develop on the wheat plant and is not affected by leaf senescence. SEM images of the adaxial side of Cranbrook and Halberd flag leaves (Figure 2A) were used to compare the stomatal density per square unit (SEM picture at 62× magnification) for the two wheat varieties. The average stomatal density of Cranbrook (72.6 ± 6.7) was lower than for Halberd flag leaves (96.7 ± 5.6; *p* < 0.05). However, the average guard cell length of Cranbrook stomata (73.6 µm ± 2.1) was higher than for Halberd (68.1 µm ± 1.4; *p* < 0.05). 

A time course experiment was established to measure stomatal conductance (SC) daily over five days of water withholding (start at T_0_). SC measurements were carried out for the top (adaxial) and bottom (abaxial) side of the flag leaf. Porometer measurements indicate that after the first day of drought treatment (T_1_) both adaxial and abaxial stomata were closed for the drought-sensitive line Cranbrook (Figure 2B). In contrast, for the tolerant line Halberd stomata were still open (adaxial) or partially open (abaxial) after the first day of drought stress (T_1_), but they were closed from the second day of drought stress onwards (T_2_; Figure 2B). SC was reduced slightly faster for the abaxial stomata for both wheat lines (Figure 2B). 

Induction of stomatal closure correlated well with induction of spike sterility in both wheat lines. When plants were re-watered after two days of drought treatment (standard drought treatment), Cranbrook spike grain numbers were reduced to 45% of unstressed control levels; after 4 days of drought treatment, grain loss was ~100% (Figure 2C). In Halberd, spike grain numbers were not affected after two days drought stress. After three and four days, Halberd spike grain numbers were reduced to 75% and 50% of the unstressed control levels, respectively (Figure 2C). The high reproducibility of the drought treatment allowed us to compare the drought response of Cranbrook and Halberd with other wheat lines. The lines CH67 and CH115 were identified as the most drought-tolerant and sensitive lines respectively of the Cranbrook × Halberd DH mapping population [53]. Porometer measurements showed that the drought-tolerant line CH67 also maintains SC up to the second day of drought treatment, while sensitive line CH115 closed stomata after the first day of drought treatment (Figure 2D). We tested the drought response of stomata for two additional drought-tolerant wheat varieties with similar drought tolerance to Halberd (data not shown) to test whether other drought-tolerant germplasm from a different genetic background behaves in the same way as Halberd and CH67. Both Westonia and Excalibur showed the same SC behaviour as the tolerant lines Halberd and CH67, closing stomata after the second day of drought treatment (Figure 2D). 

We used a similar drought stress time course experiment as described in Figure 2 to monitor the expression of the *TaNCED1* and *TaNCED2* genes during establishment of drought stress in Cranbrook and Halberd flag leaves and spikes. The expression of *TaNCED1* is induced five-fold after the first day of drought stress in Cranbrook flag leaves and expression increases further to reach maximal expression after three days (15.7-fold increase). *TaNCED1* expression then remains high after five days of drought treatment (Figure 3). In flag leaves of the tolerant line Halberd, *TaNCED1* expression is much lower and remains low throughout the five-day drought treatment, showing a 2.4-fold increase in flag leaves after two days of drought treatment (Figure 3). In Cranbrook spikes that were sampled from the same plants as the flag leaves, *TaNCED1* was induced two-fold after the first day of drought treatment and the gene was induced 11- to 14-fold from the second day onwards (Figure 3). Expression of *TaNCED2* was also strongly induced in Cranbrook flag leaves after the first day of drought treatment, but the expression level gradually decreased afterwards (Figure 3). In Halberd flag leaves *TaNCED2* expression is low, and we could not find any significant induction of the gene over the five-day treatment period (Figure 3). In Cranbrook spikes, *TaNCED2* expression was induced 1.6-fold after the first day of drought treatment and the expression increased further, peaking after three days of drought treatment (8.3-fold; Figure 3). *TaNCED2* expression in Halberd spikes showed a slight induction after two days but overall expression remained very low (Figure 3). 

The SC results indicate that reproductive stage drought-sensitive Cranbrook behaves as an isohydric variety, closing stomata immediately from the start of drought treatment, and ABA biosynthesis genes are strongly induced in both spikes and flag leaves. In contrast, the reproductive stage drought-tolerant line Halberd behaves initially like an anisohydric variety, keeping stomata open and expression of ABA synthesis genes low. However, at T_2_ Halberd tends to perform like an isohydric line; stomata also close, even though *TaNCED1* and *TaNCED2* expression levels remain low, spike sterility increases). 

### 3.3. Quantitative Differences in the Cranbrook and Halberd Drought Response

We carried out an RNAseq experiment using Cranbrook and Halberd spike and flag leaf mRNA to identify the differences in drought response between the two wheat lines. YM stage spikes and flag leaves were harvested from control unstressed plants (T_0_), and one (T_1_) and two-day (T_2_) drought-stressed plants. The entire experiment yielded a total of 30,626 differentially expressed genes (DEG). For Cranbrook we identified 23,348 DEGs, while significantly less DEGs were identified for Halberd (13,281). The overlap between the total DEGs of both lines is relatively low: 5398 DEGs (Figure 4A). The difference between the drought response of the two wheat lines becomes clearer when the response in spikes and flag leaves is separated. In flag leaves there is a large difference in the number of DEGs between drought-sensitive Cranbrook and tolerant Halberd. In Cranbrook flag leaves, 15,365 DEGs were identified for T_1_ and T_2_ combined, while in drought-tolerant Halberd there were 1309 DEGs for both time points. The overlap in flag leaf T_1_ and T_2_ DEGs between the two lines is only 528 DEGs (Figure 4A). In spikes the difference in the total amount of DEGs between Cranbrook and Halberd is less dramatic: 14,349 Cranbrook and 12,899 Halberd DEGs for the two time points combined and the overlap between the spike DEGs for both lines at T_1_ and T_2_ is again small: 3828 DEGs (Figure 4A). 

Separating the two time points of drought treatment showed that at T_1_, Halberd flag leaves and spikes had only 10 and 250 DEGs respectively. In contrast, Cranbrook showed 7342 and 3920 DEGs in flag leaves and spikes respectively (Figure 4B). At T_2_, the number of DEGs increased further in both tissues for Cranbrook. In Halberd, the number of DEGs also increased at T_2_, but the response in flag leaves remained restricted (1301 genes) compared to spikes (12,892 genes; Figure 4B). These results demonstrate a dramatic quantitative difference in drought-induced gene expression between Cranbrook and Halberd, both in flag leaves and spikes. Cranbrook is more sensitive to drought and responds from T_1_ onwards. In Halberd the drought response is delayed to T_2_; this correlates with the delay in stomatal closure we observed for this line. But the size of overlap in DEGs also indicates that there are significant qualitative differences in drought response between the two lines. Another striking difference between the two lines is the difference in spike and flag leaf response in drought tolerant Halberd at T_2_; this pattern was not observed for Cranbrook at T_1_.

### 3.4. Biological Processes Affected by Drought in Cranbrook and Halberd Spikes and Flag Leaves

GO enrichment analysis was used to identify significant functional differences in the drought response of the tolerant and sensitive wheat lines. GO enrichment analysis was first carried out using all DEGs for each line and each tissue, focusing on biological processes (BP; Table 1 and Table 2). In Halberd flag leaves, the largest enrichment of DEGs occur for the GO terms ‘regulation of cellular processes’, ‘biological processes’ and ‘transcription’ (Table 1). Many genes present in these top three GO terms overlap; they contain bZIP/G-box binding factor transcription factors (10 genes), homeodomain (12 genes) and heat stress factors (HSF; 13 genes). There are 2 differentially expressed WRKY transcription factor genes and both are repressed in Halberd flag leaves and spikes (Appendix A). The bZIP/G-box binding factors (GBF) are induced or repressed in Halberd (T_2_) and Cranbrook (T_1_ and T_2_) flag leaves and spikes, but their magnitude of response to drought is larger in Cranbrook (Appendix A). bZIP/GBF may play a role in ABA signalling. Five genes belong to the ‘photosynthesis, light reactions’ term; they are significantly enriched and up-regulated by drought in both Halberd flag leaves and spikes. The drought stress related GO terms ‘proline biosynthesis’ and ‘response to desiccation’ are enriched in both Halberd and Cranbrook flag leaves and spikes. Three potassium transporter genes were also significantly enriched (Table 1). They were repressed in both Halberd flag leaves and spikes and repressed in Cranbrook flag leaves only (Appendix A). The ‘lipid transport’ term contains 6 non-specific lipid-transfer protein (LTP) genes which are mostly drought-induced at T_2_ in Halberd flag leaves. 

In Cranbrook flag leaves, the spectrum of enriched genes was larger than in Halberd (Table 1). The top two enriched GO terms are ‘protein metabolic process’ and ‘gene expression’. The ‘protein metabolic process’ term contains several genes encoding ribosomal proteins, translation initiation factors and ribosomal proteins that play a role in mRNA translation. Interestingly, these genes are all repressed in Cranbrook flag leaves only and some are repressed in spikes as well. This suggests that translational activity is interrupted by drought stress in Cranbrook. The ‘gene expression’ class in Cranbrook also contains different types of transcription factors. Some classes of transcription factors are shared with Halberd, but there are more gene copies representing these classes: 17 bZIP/G-box binding factors, 28 homeobox and 24 HSF transcription factors (Appendix A). Enrichment of WRKY-type transcription factors is also more prevalent (38 genes, vs. 2 genes in Halberd) and many WRKY genes were induced from T_1_ onwards. Interestingly, there were also 10 genes encoding auxin-response factors (ARF; seven induced, three repressed). Only two of these ARFs were also expressed in Halberd (Appendix A). There were 8 enriched MADS-box transcription factors in Cranbrook flag leaves; seven were induced at T_2_, while one was repressed at T_1_. MADS-box factors play a role in reproductive development and were not significantly enriched in drought-stressed Halberd flag leaves. Three wheat phytochrome A (PhyA) genes were present amongst the ‘gene expression’ GO terms. Two of these genes were induced from T_1_ onwards in Cranbrook and from T_2_ in Halberd (Appendix A). The ‘fatty acid biosynthetic process’ term is also enriched in Cranbrook flag leaves. This term includes 29 3-ketoacyl-CoA synthase and 13 fatty acid hydroxylase genes which may be involved in cuticular wax synthesis.

In Halberd spikes, the GO term ‘response to stimulus’ contained the largest amount of DEGs (Table 2). This class contains several known stress responsive genes encoding ABA-stress-ripening, Late Embryogenesis Abundant (LEA) and Universal Stress Proteins (USP). Most of these genes are predominantly induced in Halberd spikes (Appendix A). An important set of genes encoding proteins involved in protection against Reactive Oxygen Species (ROS) is also represented in this GO term: catalase (Four genes), glutathione peroxidase (2), peroxidase (76), thioredoxin (3). The presence of ‘lipid biosynthetic process’ GO term is represented by 59 genes, indicating that changes in membrane lipids are important in drought-stressed Halberd spikes. The ‘response to desiccation’ and ‘response to stimulus’ GO terms include 6 LEA genes, while the microtubule and cell cycle related GO terms contained several genes (e.g., kinesin) that were repressed at T_2_ in Halberd spikes (Appendix A). The ‘lipid biosynthetic process’ term consists of 59 genes involved in lipid (3-oxoacyl-synthase, acyl-desaturase) and cuticular wax synthesis (3-ketoacyl-CoA synthase, fatty acid hydroxylase). 

‘Auxin response’ and ‘cytokinin metabolic process’ GO terms are enriched in the Halberd spike BP GO terms only (Table 2). The ‘auxin response’ term is represented by 37 genes of the Small Auxin Up-Regulated (SAUR) gene family, while the ‘cytokinin metabolic process’ term contains 12 cytokinin dehydrogenase (CkDH) genes (Appendix A). Most of the SAUR and cytokinin dehydrogenase genes are induced at T_2_ in Halberd spikes and only one is also induced at T_2_ in the flag leaves. The same genes were repressed in Cranbrook flag leaves at T_1_ and/or T_2_ and they were not differentially expressed in Cranbrook spikes.

In Cranbrook spikes, the three dominant GO terms relate to localisation and transport, including transporters and permeases, as well as aquaporin genes that are responsible for movement of water across membranes (Table 2). Other enriched GO terms relate to photosynthesis and chlorophyll synthesis: ‘photosynthesis’, ‘tetrapyrrole biosynthetic process’ and ‘pigment biosynthetic process’ (Appendix A). Again, the genes included in these terms are mostly repressed at T_2_ in Cranbrook spike and flag leaves, indicating that photosynthetic activity is arrested by drought stress in the sensitive line. The timing of this event correlates with the timing of stomatal closure in Cranbrook flag leaves from T_1_ onwards (Figure 2B). In terms of protection against ROS, the response of Cranbrook is less substantial than Halberd, with induction of only 12 thioredoxin genes (GO term ‘cell redox homeostasis’). 

The BP GO enrichment analysis reveals five important differences in the drought response of Halberd and Cranbrook. Firstly, Halberd maintains growth by keeping stomata open and mounts a stronger stress defence response to achieve this: induction of photosynthesis genes, stronger ROS defence, induction of LEA and USP genes. But at T_2_, cell division processes in Halberd spikes are starting to be affected by drought as well - which agrees with the onset of stomatal closure. Secondly, Cranbrook is entering in growth stagnation mode: repression of transcriptional/translational activity and photosynthesis, weaker response to ROS compared to Halberd. Thirdly, while the response in Cranbrook spikes and flag leaves starts at T_1_, in Halberd most responses are activated from T_2_ onwards. This shows that Cranbrook is more sensitive to water stress, which agrees with the earlier stomatal closure we observed for this line compared to Halberd (Figure 2). Interestingly, the response at T_2_ in Halberd is very different from the response at T_1_ in Cranbrook. Fourthly, modifications in membrane lipid synthesis to maintain membrane fluidity, and cuticular wax deposition to seal cell walls to prevent water loss, occur in both wheat lines. However, in Cranbrook the response is confined to the flag leaves, while in Halberd the response is more substantial and occurs mainly in the spike. Fifthly, the presence of ‘auxin response’ and ‘cytokinin metabolism’ enriched GO terms at T_2_ in Halberd spikes only indicates that these two growth hormones may be involved in controlling the stress tolerance phenotype of this wheat line.

### 3.5. Differences in Spike and Flag Leaf Drought Stress Response between Cranbrook and Halberd 

We carried out a GO enrichment analysis for Molecular Functions (MF) using the ‘total’ and ‘specific’ (= not including overlapping genes) DEGs for each wheat variety. The results of the MF GO term analyses are listed in Appendix A. The analysis revealed a large amount of molecular and physiological processes that are affected by drought stress. Obvious observations are the enrichment of a large amount of protein kinase (receptor kinase, Ser/Thr kinase) and disease resistance genes that are specifically induced by drought stress in Cranbrook flag leaves (‘anion binding’ GO term). The prevalence of disease resistance genes in the response to drought illustrates that the role of these genes extends beyond pathogen defence. In Halberd flag leaves and spikes, a large amount of P450-type oxidoreductase enzymes are enriched (‘oxidoreductase activity’, ‘heme binding’ GO terms). P450 enzymes play a role in many metabolic processes, including hormone metabolism. In Cranbrook spikes, genes encoding P450 family enzymes are also enriched, but there is a substantial difference in the genes enriched in Halberd and Cranbrook spikes (Appendix A). Another striking difference in the drought response between the two wheat lines is the enrichment of many genes encoding histone superfamily proteins (‘DNA binding’ GO term) in Halberd spikes; most of these chromatin modification genes are repressed in Halberd only.

The drought response in Halberd is also characterised by enrichment of genes encoding the peroxidase oxidative stress response enzyme in flag leaves (9 genes) and spikes (73 genes), as well as 3 genes encoding catalase in spikes (‘heme binding’ GO term). This may indicate that the tolerant line has a better ROS defence response compared to sensitive line Cranbrook. In Halberd, enrichment for genes involved in lipid metabolism and cuticular wax/cuticle deposition is also more dominant compared to Cranbrook. 11 (7 repressed and 4 induced) acyl-[acyl-carrier-protein] 6-desaturase genes are enriched at T_2_ in Halberd spikes only (‘oxidoreductase activity acting on the aldehyde or oxo group of donors, NAD or NADP as acceptor’ GO term; Appendix A). Enrichment for genes involved in long-chain fatty acid production for cuticular wax deposition include 7 fatty acyl-CoA reductases (‘oxidoreductase activity’ GO term) induced in Halberd spikes, 4 and 12 fatty acyl-CoA reductases in Halberd and Cranbrook spikes respectively (‘oxidoreductase activity’ GO term) and 7 fatty acid hydroxylases in Halberd flag leaves (‘oxidoreductase activity’ GO term). The fatty acyl-CoA reductase genes in Cranbrook are repressed in the spike and induced in the flag leaf. Three classes of genes involved in lipid transport, lipid homeostasis and cuticle deposition in the membrane and cell wall space are also enriched. Several GDSL esterase/lipase genes (‘hydrolase activity’, ‘catalytic activity’ GO terms) are specifically enriched in Halberd (83 genes) and Cranbrook (48 genes) spikes. While in Halberd these genes are mainly induced/repressed by drought in the spike, in Cranbrook some genes are also expressed in the flag leaf. Genes encoding HXXXD-type acyl-transferase family proteins are enriched in Halberd spikes (12) and Cranbrook flag leaves (73 genes). 19 ABC transporter genes (‘transmembrane transporter activity’) are enriched in Cranbrook flag leaves and only four of these genes are induced by drought in Halberd spikes (Appendix A). Although we observed a functional overlap in lipid responses in the two wheat varieties, the GDSL esterase/lipase and HXXXD-type acyl-transferase gene copies recruited by both varieties are different. This may indicate that these genes respond to different regulatory cues in the tolerant and sensitive wheat lines. It is possible that the use of different lipid metabolism genes in both wheat lines may lead to differences in membrane composition and fluidity, as well as differences in water permeability of the cuticle in both wheat lines.

### 3.6. Identification of Drought Responsive CBF/DREB Factors in Halberd and Cranbrook

Many classes of transcription factors (ERF, Myb, Zinc finger factors, ERF, bHLH) are enriched in Halberd and Cranbrook spikes and flag leaves. Of special interest for response to drought stress are CBF/DREB transcription factors. The MF GO enrichment analysis identified a total of 6 enriched CBF/DREB genes and the expression of these genes is quite different in the two wheat lines and the two tissue types. four genes are significantly enriched in Halberd spikes and 5 in Cranbrook flag leaves (‘DNA-binding transcription factor activity’ term); the expression pattern of these genes is shown in Figure 5A. Three of these genes are shared by the two wheat lines, but in Cranbrook they are induced at T_2_ in flag leaves, while in Halberd they are expressed at T_2_ in the spike. One gene is only expressed at T_2_ in Halberd spikes, while another two genes are only expressed in Cranbrook flag leaves (induced at T_1_, repressed at T_2_) and spikes (induced at T_2_). The total DEG list contained an additional 6 CBF/DREB-like genes (total of 12 genes; Appendix A). We tested the expression of two genes (one significantly enriched and one non-enriched gene) using RNA from a new drought treatment experiment (Figure 5B). The TRIAE_CS42_4AL_TGACv1_290565_AA0986890.1 gene showed induction by drought stress in Cranbrook flag leaves, but not in spikes - where the gene is repressed. This CBF/DREB gene is also repressed in Halberd flag leaves and spikes. The TRIAE_CS42_2AL_TGACv1_093266_AA0276100.1 gene is strongly induced by drought in Cranbrook flag leaves but is repressed in spikes. In Halberd this gene is again repressed in both flag leaves and spikes (Figure 5B). Although expression levels of this gene were similar in Cranbrook and Halberd flag leaves, expression was much higher in Halberd spikes at T_0_ (21-fold) and remained higher than Cranbrook at T_2_ (6.3-fold). These results indicate that CBF/DREB genes are expressed quite differently in flag leaves and spikes of the drought-tolerant and sensitive wheat lines used in this study.

### 3.7. Differential Expression of ABA-Related Genes in Drought-Stressed Cranbrook and Halberd

The MF analysis revealed that a total of eight *TaNCED* genes were enriched in spikes and flag leaves of both wheat lines. All eight genes were enriched in Halberd spikes only, while six where enriched in both spikes and flag leaves of Cranbrook (Figure 6A). While six *TaNCED* genes were expressed in both Halberd and Cranbrook spikes, two genes were specific for Halberd spikes (see below). *TaNCED2* is one of the enriched genes (asterisk in Figure 6A), but the closest match for the *TaNCED1* gene (see Materials and Methods) was not present in the DEG list. In Cranbrook flag leaves, three *TaNCED* genes (including *TaNCED2*) were induced after one day of drought stress, while six *TaNCED* genes were induced after the second day of drought treatment (Figure 6A). In contrast, Halberd flag leaves did not show any significant induction for any of the *TaNCED* genes at both time points. In Cranbrook spikes, four *TaNCED* genes were already induced by drought stress after one day and six genes were induced after the second day of treatment (Figure 6A). All eight *TaNCED* genes were induced in Halberd spikes, but only after the second day of drought stress. The two *TaNCED* genes that were induced by drought at T_2_ in Halberd spikes only may encode the closely related carotenoid cleavage dioxygenase enzyme (CCD). Overall induction levels of *TaNCED* genes in Halberd spikes are lower than in Cranbrook spikes and flag leaves (Figure 6A). 

We carried out ABA measurements in flag leaves and spikes after a two-day drought stress treatment (T_2_; Figure 6B). The results show that ABA levels increase significantly in drought-stressed Cranbrook flag leaves and spikes as compared to control. Despite the induction of *TaNCED* genes in Halberd spikes at T_2_, ABA levels in spikes were found to be reduced by drought stress. In contrast, in Halberd flag leaves the ABA level increased ~3.5-fold (Figure 6B), even though *TaNCED* genes are not significantly induced (Figure 6A). Drought-tolerant line Halberd is somehow able to repress ABA accumulation in the spike, possibly through fast catabolism or mobilisation of the hormone. ABA accumulation in the flag leaf may be the result of transport from the roots or the spike, or recruitment from stored ABA conjugates. The lower levels of ABA in Halberd flag leaves may be sufficient to initiate stomatal closure after two days of drought treatment. It is possible that ABA levels in Halberd flag leaves reach higher levels during prolonged drought treatment (>2 days). Two other potential ABA biosynthetic genes were identified in the DEG list. In Cranbrook, three genes encoding short chain dehydrogenase/reductases (*SDR*), enzymes producing carotenoid precursors for ABA biosynthesis [59], were induced at T_2_ in flag leaves and at T_1_ in spikes (Figure 6A). These genes were not differentially expressed in Halberd. In Halberd, two genes potentially encoding zeaxanthin epoxidase, the first committed step in ABA biosynthesis from carotenoid precursors, were drought-induced in spikes at T_2_ (Figure 6A). 

Four genes encoding protein phosphatase 2C (PP2C; ‘catalytic activity’ term) were significantly enriched in Cranbrook flag leaves, as well as Halberd spikes and flag leaves (Appendix A). The two genes enriched in Halberd are repressed by drought stress, while one gene is induced at T_1_ and T_2_ in Cranbrook flag leaves. Many more potential PP2C genes are amongst the DEG and they are differentially expressed in both wheat varieties (Appendix A). Members of the PP2C gene family are important components of the ABA signalling pathway and control proline and osmolyte accumulation under drought stress [60]. 23 bZIP transcription factors and potential G-box binding factors were also identified using the GO term enrichment analysis. Eleven of these genes were DEGs in Cranbrook only and most are induced or repressed from T_2_ onwards and mainly in flag leaves; 5 genes are repressed and 5 induced in Halberd flag leaves (Appendix A). We confirmed the expression pattern for one of these bZIP transcription factors (TRIAE_CS42_1AL_ TGACv1_001758_AA0034810.2; Figure 6C) to illustrate the difference in expression of this gene in Halberd and Cranbrook flag leaves and spikes.

### 3.8. Potential Role of Auxin in the Drought Response of Halberd and Cranbrook

There were 37 SAUR-family genes that were enriched in the ‘auxin response’ GO term in drought-stressed Halberd spikes (Table 2; Appendix A). The vast majority of these SAUR genes were induced at T_2_ and 25 of them were only expressed in Halberd spikes. 14 SAUR genes were also expressed in Cranbrook flag leaves, but they were all repressed at T_1_ or T_2_ (Appendix A). The expression of one of the SAUR genes (TRIAE_CS42_5AL_TGACv1_374113_AA1190830.1) was confirmed by real-time PCR (Figure 7A). 13 Auxin response factor (ARF) genes were also significantly enriched (Appendix A). Interestingly, two of these genes were repressed at T_2_ in Halberd spikes only. The other ARF genes were induced from T_1_ onwards (5 genes) or repressed at T_2_ (3 genes) in Cranbrook flag leaves. The MF GO enrichment analysis also identified three ARF genes (‘DNA binding’ GO term) and they were all repressed at T_2_ in Halberd spikes (Figure 7B). Furthermore, the MF analysis identified 18 potential flavin monooxygenase genes (FMO; ‘flavin adenine dinucleotide binding’, ‘anion binding’ GO terms; Figure 7B). Tryptophan aminotransferase (TAA, YUCCA), which catalyses the rate-limiting step in indole-3-acetic acid (IAA) biosynthesis, is an FMO family enzyme. Twelve of the potential YUCCA encoding genes (four down- and eight up-regulated) are specifically expressed at T_2_ in Halberd spikes, while another four genes are repressed in Cranbrook flag leaves only. Another two genes are induced at T_1_ and T_2_ in Halberd and Cranbrook spikes respectively (Figure 7B). The expression of a chromosome 5A FMO gene was confirmed by real-time PCR. The expression is repressed in spikes of both wheat lines, but this gene is induced in Cranbrook spikes (Figure 7C). 

We also determined levels of the phytohormone auxin by measuring indole-3-acetic acid (IAA) and the auxin precursor indole-3-butyric acid (IBA) and the auxin derivative methylated IAA (MeIAA) in spikes and flag leaves of the two wheat lines (two-days drought stress; Figure 8). The results show that in spikes IAA levels are decreased by drought stress in Halberd, but they increased in Cranbrook. This result agrees with the expression of some of the FMO genes (Figure 7). In the flag leaf, Halberd IAA levels are induced slightly but in Cranbrook flag leaves IAA levels are strongly increased by drought treatment (Figure 8). We also measured the IAA precursor indole-3-butyric acid (IBA), the accumulation of which may stimulate IAA accumulation. IBA was not detectable in spikes of both Halberd and Cranbrook, but its levels significantly increased in flag leaves of both wheat lines (Figure 8). MeIAA is involved in controlling polar transport of IAA and coordinating plant development [61]. MeIAA levels were not detectable in unstressed Halberd spikes, but its levels were strongly induced by drought stress. This may explain why auxin levels do not increase in Halberd spikes, as the hormone is relocated. In Cranbrook, spike MeIAA levels were already high under unstressed conditions and levels further increased upon drought treatment (Figure 8). In flag leaves, MeIAA levels were not detected in unstressed conditions, but levels increased under drought conditions in both Halberd and especially Cranbrook (Figure 8). The MeIAA level differences may point to differences in polar transport of auxin between Halberd and Cranbrook. Amongst the total DEGs are 17 genes encoding potential auxin efflux carrier proteins (Appendix A). 11 of them are repressed at T_2_ in Halberd spikes (8 genes Halberd specific) and 1 gene is induced by drought stress. Three of the Halberd repressed genes are also repressed in Cranbrook spikes (and induced in flag leaves), while three different genes are repressed at T_2_ in Cranbrook spikes only (Appendix A). 

Further analysis of the complete DEG list revealed several other auxin-related genes (Appendix A). These genes include GH3 family members involved in conjugation of auxin to various amino acids, and Aux/IAA transcriptional repressors of auxin signalling. 11 Auxin-responsive GH3 family proteins are differentially expressed; 2 were repressed in Cranbrook spikes and flag leaves, 6 were induced in Halberd spikes only, and another three genes were repressed at T_2_ in both Halberd and Cranbrook spikes and induced in Cranbrook flag leaves. Taken together, the differential expression of auxin-related genes indicates that regulation of auxin homeostasis and translocation of auxin differs significantly between Halberd and Cranbrook.

Amongst the many F-box protein genes in the total DEG list, 8 showed homology to auxin F-box proteins (Appendix A). 8 of these genes are induced and three are repressed at T_2_ in Cranbrook flag leaves; none of these genes are differentially expressed in Halberd. 7 genes encoding SKP1-like proteins are repressed by drought in Cranbrook flag leaves and spikes only. SKP1 is together with F-box proteins part of the SCF proteasome complex that initiates auxin signalling. These data demonstrate that the tolerant and sensitive wheat lines also differ in auxin-mediated signalling under drought conditions.

### 3.9. Role of Cytokinin in the Drought Response of Halberd and Cranbrook Spike and Flag Leaves

Reference to a role for cytokinins in the drought response came from both the BP (‘cytokinin metabolic process’) and MF (‘flavin adenine dinucleotide binding’, ‘anion binding’) GO terms appearing in the GO enrichment analyses (Table 2; Appendix A). Enrichment for these terms occurred in Halberd flag leaves and spikes, as well as in Cranbrook flag leaves. The genes included in these GO terms are 15 genes encoding CkDH, an enzyme involved in cytokinin catabolism (Figure 9A). 5 of these genes are specifically induced at T_2_ in Halberd spikes, and another four genes are induced and 1 is repressed at T_1_ in Cranbrook flag leaves. Another 2 *TaCkDH* genes are only induced by drought stress at T_1_ in Cranbrook flag leaves, while three genes are repressed at T_2_ in both Halberd flag leaves and spikes (Figure 9A). We confirmed the expression for one of the *TaCkDH* gene (TRIAE_CS42_3B_TGACv1_225183_AA0805930.1) by real-time PCR (Figure 9B). The expression of this gene peaks at T_1_ in both Halberd spikes and Cranbrook flag leaves and returns to normal levels at T_2_. This indicates that cytokinin homeostasis may differ in the drought-stressed tolerant and sensitive wheat lines. Levels of the cytokinin zeatin are significantly induced by drought in both flag leaves and spikes of Cranbrook, but not in Halberd (Figure 9C). In Halberd, induction at T_2_ of the *TaCkDH* genes may be responsible for the lower zeatin levels in spike and flag leaves (Figure 9B).

The BP GO enrichment analysis also revealed 7 genes encoding type A response regulators (Appendix A). All but two of these genes are repressed at T_2_ in Halberd spikes and flag leaves, but the same genes are also repressed in Cranbrook flag leaves (T_2_) and spikes (T_1_ and T_2_; Appendix A). Type-A response regulators are two-component signalling factors that play a role in signalling of different plant hormones, including cytokinin, auxin and ethylene. 

### 3.10. Effect of Drought on Ethylene Responsive Genes in Halberd and Cranbrook

The fourth hormone we identified using the GO enrichment analysis is ethylene. The MF GO term ‘DNA-binding transcription factor activity’ contains 105 ethylene response factors (ERF) genes that are enriched in Halberd spikes and Cranbrook flag leaves (Appendix A). 36 ERF factors were uniquely expressed in Cranbrook and 32 are specifically expressed in Halberd spikes. In Halberd, ERF genes are mainly expressed in the spike at T_2_ (Appendix A). Three enriched ethylene receptor genes were induced by drought at T_1_ and T_2_ in Cranbrook spikes only (Appendix A). There were no ethylene biosynthesis genes present in the GO enrichment analysis, but the total drought-responsive DEG list contained twenty-four potential candidates for the ethylene biosynthetic enzyme 1-aminocyclopropane-1-carboxylate oxidase (ACC oxidase, ACO; Appendix A). Apart from one gene (repressed at T_2_ in flag leaves), all of these genes were induced in both flag leaves and spikes of the drought-sensitive variety Cranbrook. Only seven genes were also expressed in Halberd flag leaves and spikes, but from T_2_ onwards. Three of these genes were repressed in both spikes and flag leaves, while the other two were only induced in spikes (Appendix A). This may indicate that ethylene may play a more dominant role in the drought-sensitive variety Cranbrook.

### 3.11. The Drought Response at T_1_ in Halberd Flag Leaves

A surprising observation was the restricted drought response in Halberd flag leaves after only one day of drought treatment. At this stage, stomata are still open and only 10 genes were found to be differentially expressed (Appendix A). All 10 genes are expressed at T_1_ and in flag leaves only and three of these genes are also expressed at T_1_ in Cranbrook flag leaves, albeit with an opposite regulation pattern. The functional annotations for these 10 genes could not be identified unequivocally using BLAST searches with the rice and barley annotations. Three of these 10 genes have a completely unknown function (‘expressed protein’). Another three genes show some similarity to LTPL4-type protease inhibitors; they are induced in Halberd flag leaves only. Another gene shows similarity to a Bowman-Birk type protease inhibitor; this gene is repressed in Halberd flag leaves alone. Two other genes show similarity to early nodulation (ENOD) genes and are repressed in Halberd flag leaves. One RGA2-like disease resistance gene was induced in Cranbrook and repressed in Halberd flag leaves. Another gene encoding an unknown protein is only repressed in Halberd flag leaves (Appendix A). Although the identity and function of the proteins these 10 genes encode remains to be established, but they point to the possibility that the early drought response in Halberd flag leaves may not be triggered at the transcriptional level but at the post-translational level.

## 4. Discussion

Land plants evolved from aquatic photosynthetic green algae (charophytes) about 475 million years ago [62,63]. Early land plants lacked the ability to regulate their water balance. During periods of water shortage, desiccation tolerance (DT) allowed them to survive in an inactive, dehydrated state. DT still exists in some land plants (bryophytes, resurrection plants), but in vascular plants DT is restricted to reproductive structures that require dispersal in the environment (pollen grains, seeds) [64,65]. Higher plants evolved roots, vascular tissues and stomata to actively acquire water and nutrients from the soil and this was instrumental for conquering the Earth’s land mass [66,67,68]. These morphological changes required the need for angiosperms to regulate their water balance and ABA was recruited to adjust stomatal conductance and regulate uptake/transport of water [39,69]. Genetic differences in the capacity to regulate water balance, transpiration and stomatal conductance is therefore expected to affect the plant’s capacity to grow and adapt to environments with limited water availability [35]. In wheat, water use efficiency (WUE) and transpiration efficiency is not strongly correlated with improvements in grain yield under drought conditions [46,70,71,72]. These traits improve growth and biomass production during vegetative-stage drought stress, but they do not necessarily lead to higher grain yields when water stress conditions persist during the reproductive stage [8,46,71,73,74]. Although osmotic adjustment is difficult to phenotype, osmotic adjusting lines have been shown to support higher grain yields in wheat [75,76,77,78].

### 4.1. Active and Passive Survival of Drought Conditions

Higher plants acquired both passive and active survival mechanisms to survive long and short-term drought stress. Isohydric plants (‘water savers’) close stomata to reduce transpiration and improve WUE. Anisohydric plants (‘water wasters’) take risks by maintaining stomatal conductance and transpiration [34]. Instead, they increase osmotic potential to maintain cellular water levels [9,79]. Osmotic adjustment can temporarily overcome water stress conditions, but survival is compromised when water availability in the environment becomes seriously depleted. Evidence is emerging that isohydry and anisohydry are not purely species-specific, nor are they environment-specific adaptation mechanisms [35,80]. Our results confirm this hypothesis by showing that wheat can be either isohydric or anisohydric depending on the length/severity of drought conditions. Furthermore, our data show that there is genetic diversity in wheat for the threshold level of drought stress that causes plants to switch from anisohydry to isohydry and passive survival mode. Our results imply that increasing the threshold level of drought stress at which wheat plants close stomata during reproductive development may improve grain set. It is therefore important to gain an understanding of how this threshold is determined at the molecular level.

In our controlled environment (CE) drought assay, the drought sensitive and drought tolerant varieties close stomata after one and two days of drought treatment, respectively. All drought-tolerant lines we tested showed the same response, closing stomata after two days of drought treatment. This may mean that our drought assay does not have the resolution to pick up differences in SC behavior and drought-tolerance between the lines we tested. As anisohydry is a short-term survival mode, it is also likely that there may not be much potential to increase the level of drought stress severity at which stomata close. Drought experiments using pot plants impose severe restrictions on root growth and the constant ventilation of the growth cabinet can quickly establish severe drought conditions. In the field, drought conditions are established much more gradually compared to our controlled environment assays, so anisohydric wheat lines will maintain higher stomatal conductance for a longer period of time. Photosynthesis and sugar transport to the spike will therefore also last longer during active spike development, enabling anisohydric lines to ultimately develop more grains. Entering passive survival during the metabolically active stage of spike development and yield determination is more likely to have negative consequences in terms of grain yield. Many abiotic stresses affect stomatal conductance. It remains to be established whether the genetic diversity for drought tolerance we describe here may also benefit tolerance to heat and cold stress. 

### 4.2. The Role of ABA in Reproductive Stage Drought Tolerance

Previous evidence has shown that ABA accumulation at the YM stage of pollen development represses sugar delivery and sink strength in drought stressed wheat anthers by directly/indirectly repressing cell wall invertase expression in the tapetal cell layer. Sugar delivery to the tapetum is arrested, causing abortion of pollen development and sterility [45,49]. At the YM stage, anthers have the highest sink strength in the spike and cross-pollination experiments have shown that the ovary is more resilient to drought stress [49]. In vegetative plant parts, induction of ABA synthesis by water stress causes stomatal closure and reduced transpiration [46,69,81]. ABA and sugar response pathways are known to interact [82,83] and ABA inhibits expression of photosynthesis genes [84,85]. The effects of ABA at the vegetative and reproductive level therefore negatively affect plant growth and development. Although improvement of WUE and transpiration efficiency are beneficial for plants to overcome vegetative-stage water stress [46,48,70,71,72], this approach to restrict water loss is likely to induce a passive growth response. During reproductive development this will affect grain yield in wheat. Transcriptome profiling identified 8 potential *TaNCED* genes that are strongly induced in Cranbrook flag leaves and spikes, but induction of these genes in Halberd is restricted and ABA levels in the flag leaf at T_2_ are much lower than in Cranbrook, despite the low induction levels of *TaNCED* genes. This indicates that mobilisation of ABA from other sources (roots, spike), or recruitment from inactive conjugation products plays an active role in controlling ABA levels. ABA is known to be transported from the site of synthesis in vascular tissues (*TaNCED*) to the site of action [45,86]. The regulation of ABA transport is not fully understood, but our results show that during flowering in drought-tolerant Halberd this process may be partially driven by signals from the reproductive organs. ABA is not only synthesized in the roots but at multiple sites in the plant, resulting in complex signalling networks [87]. Although ABA may be involved in determining both vegetative and reproductive drought tolerance, the control of ABA homeostasis in both plant parts may be independent and under the control of different upstream regulators. 

### 4.3. Halberd and Cranbrook Have a Different Response to Drought Stress 

The transcriptome analysis revealed several differences in the drought response of Cranbrook and Halberd. Overall, the tolerant line Halberd mounts a stronger defence response to drought stress than Cranbrook. This protective response is regulated by factors other than ABA; ABA levels in Halberd are lower than in Cranbrook - particularly in spikes. The tolerant line Halberd induces genes involved in photosynthetic light reactions, several LEA and USP-type proteins and it activates a stronger ROS response (catalase, glutathione peroxidase, peroxidase, thioredoxins) in the metabolically active spike. Protection of macromolecules is essential under all abiotic stresses to protect against oxidative stress [88]. However, signs that microtubule and cell cycle genes are repressed at T_2_ in Halberd spikes when stomata are starting to close indicate that Halberd is starting to experience drought stress. Cranbrook’s response to drought in flag leaves is repression of ribosomal and RNA translational protein synthesis, as well as photosynthesis related proteins in the spike. The overall response to ROS is also much weaker in Cranbrook. These responses indicate that Cranbrook is repressing growth and preparing for longer-term drought stress by entering a passive growth mode, while Halberd is initially maintaining growth. This requires a stronger drought defence response to protect macromolecules and maintain cellular activities.

Protection of cellular membranes is another important stress response mechanism [89]. Enrichment for GO terms involved in lipid biosynthesis and transport occur in both Halberd and Cranbrook flag leaves and spikes, but there are significant quantitative and qualitative differences in the genes that are affected by drought stress in both wheat lines. Drought stress appears to affect membrane composition and activate the deposition of cuticular wax in both wheat lines. The cuticle is a protective hydrophobic layer consisting of high chain length fatty acids deposited between the plasma membrane and cell wall that prevents water loss [90,91,92]. The cuticle structure is complex and can vary widely in composition. It can play a role in regulating water loss, osmotic stress response and ABA biosynthesis [93,94]. It is not clear to what extend the drought-induced differences in gene expression we observed could cause differences in cuticle composition in Halberd and Cranbrook and how this could affect tissue water holding capacity between the tolerant and sensitive wheat line. 

CBF/DREB transcription factors play an important role in establishing an acclimation response to drought and other abiotic stresses [95,96]. 6 CBF/DREB TF genes were enriched in Halberd and Cranbrook spikes and flag leaves respectively, and an additional 6 genes were identified in the total DEG list. Confirmation experiments showed that these genes were mainly induced in Cranbrook flag leaves and but not in spikes of both wheat lines. All CBF/DREB genes are expressed quite differently in flag leaves and developing spikes of the drought-tolerant and sensitive wheat lines used in this study. Despite the fact that overexpression of CBF/DREB TF under the control of different promoters has frequently been shown to improve drought tolerance in various crop plants [97,98,99], the mechanism used by the variety Halberd to maintain higher spike grain number under drought conditions is likely due to involve other regulatory factors. 

### 4.4. Drought Stress Differentially Affects Auxin and Cytokinin Metabolism and Signalling

The GO enrichment analysis clearly revealed that two other hormones play a role in controlling reproductive drought tolerance: auxin and cytokinin. It is possible that these growth hormones are responsible for the observed differences in ABA behaviour in the tolerant and sensitive wheat line. ABA and auxin interact to control root hair elongation [100,101] and the GH3 auxin conjugation enzyme can modulate auxin and ABA homeostasis and affect drought tolerance [102]. In Arabidopsis, the *dnd2* mutant was shown to increase auxin and ABA levels, resulting in reduced stomatal conductance [103]. The ‘auxin response’ genes consist mainly of SAUR genes, but several other DEGs controlling auxin homeostasis (YUCCA, GH3) and signalling genes (ARF, AUX/IAA, auxin efflux carriers) were also expressed differently in the two drought-stressed wheat lines. The only cytokinin metabolism genes that were expressed differently in the two wheat lines were cytokinin dehydrogenase genes, which are involved in cytokinin catabolism [104]. Most of the YUCCA-like genes are induced at T_2_ in tolerant line Halberd. 6 out of 9 auxin-inducible GH3-like genes that conjugate auxin to amino acids [105,106] are up-regulated at T_2_ in Halberd. These gene expression changes show that drought stress affects auxin homeostasis. Measurements of auxin levels show that the hormone is only increased by drought stress in Cranbrook flag leaves and spikes. Most cytokinin dehydrogenase genes are induced at T_2_ in Halberd spikes and at T_1_ in Cranbrook flag leaves. From these gene expression changes we expected cytokinin levels in Halberd spikes to be reduced and this is what we observed. In contrast, cytokinin levels increased in both Cranbrook flag leaves and spikes. Cytokinins also play a role in drought stress via interactions with ABA and auxins [107]. There is ample evidence for auxin-cytokinin cross talk and the balance between both hormones is important for regulating plant growth and development [108,109]. Auxins are normally synthesised in the shoot apex, but synthesis moves to the floral meristem during initiation of flowering in wheat and later to the developing spike [110]. In wheat, auxin signalling increases and cytokinin signalling decreases during spike development [111]. The physiological effect of both auxin and cytokinin is difficult to estimate and depends on their steady state levels, transport and signalling activities, as well as interactions with other hormones. In recent years, several reports have implicated auxin metabolism and signalling in drought-tolerance in plants [102,112,113,114]. It has been known for quite some time that auxin is involved in controlling turgor, cell division and extension and osmoregulation [115,116,117,118,119]. Although the role of SAUR proteins has been elusive for a long time, recent data show that they function as mediators of hormonal response and regulators of plant growth in function of environmental signals [120]. SAUR proteins have recently been shown to interact with PP2C and activate plasma membrane (H^+^)-ATPases to regulate cell expansion [121,122]. There is a large amount of SAUR genes that are differentially expressed under drought conditions in Halberd spikes, but some of these genes could also play a role in other hormone responses, including ethylene, jasmonate and cytokinin [123,124]. Some SAUR genes act as negative regulators of auxin synthesis and transport [125]. 

### 4.5. Role of Ethylene in Controlling Reproductive Drought Tolerance

GO enrichment analysis also revealed several differentially expressed AP2/ERF transcription factors that play a role in stress responses [126], as well as three ethylene receptor-like genes that were uniquely induced in Cranbrook spikes. The CBF/DREB factors also belong to the same TF family [95,96]. Expression of ACC oxidase-like ethylene biosynthetic genes is induced by drought stress in Cranbrook flag leaves and spikes, while only few of them were differentially expressed in Halberd (three genes were repressed). Ethylene is also a regulator of stomatal conductance and is involved in senescence responses [81,127]. In addition, ethylene, ABA, auxin and cytokinin have been shown to have agonistic and antagonistic interactions [128,129]. Both cytokinin and auxin have been shown to promote ethylene production and ABA-induced stomatal closure [81,130]. 

### 4.6. What Is Happening at T_1_ in Halberd Flag Leaves?

After the first day of drought treatment, only 10 DEGs were detected in Halberd flag leaves compared to 7342 DEGs in Cranbrook. Even after 2 days, the response in Halberd flag leaves (1301 DEGs) is much smaller compared to Cranbrook (12,892 DEGs). The small number of genes is somewhat expected. Unlike Cranbrook, Halberd flag leaves have not yet responded to drought stress by stomatal closure. Is Halberd simply more insensitive to drought stress, or is Cranbrook over-sensitive to drought stress? According to their expression pattern, the 10 genes can be divided in three classes. Three homologous protease inhibitor/seed storage/LTP protein genes are up-regulated only in Halberd flag leaves. Two early-nodulation and one Bowman-Birk-type trypsin inhibitor gene are repressed only in Halberd flag leaves. Bowman-Birk type trypsin inhibitors have previously been implicated in salt and drought tolerance through regulation of stomatal conductance and relative water content, as well as reduction in oxidative stress levels [131,132]. The third group of genes are down-regulated at T_1_ in Halberd flag leaves, but are induced in Cranbrook flag leaves. This group of four genes consists of three genes with unknown functions and one RGA2-like disease resistance gene. NBS-LRR disease resistance genes have also been implicated in drought tolerance [133]. The exact function of these genes in the early drought response of the drought tolerant line Halberd is unclear and needs to be elucidated by further experimentation.

## 5. Conclusions

Drought stress is one of the more common abiotic stresses to affect productivity of crop plants such as wheat. The results presented in this paper show that wheat plants have an in-built mechanism to adapt to drought stress for the short term (anisohydry) or the long term (isohydry). Importantly, there is genetic variation that determines at which threshold level of drought stress the short or long-term adaptation mechanism is activated. In addition, we show that the more conservative isohydric response to drought has disadvantages during the reproductive stage: spike development is arrested, affecting grain yield. In contrast, wheat plants with higher drought tolerance during the reproductive stage can delay the isohydric response until a certain threshold level of drought stress (anisohydry) before becoming isohydric. We show that this has a positive effect on grain yield, most likely because spike development can continue longer compared to isohydric lines. These findings may lead to selection strategies to increase the threshold level of drought stress at which wheat plants switch from anisohydry to isohydry. Although, it is uncertain how far this limit can be pushed. The transcriptome analysis produced a large amount of information showing that there are drastic differences in the drought response of the sensitive and tolerant wheat lines. The results show that interaction between plant hormones (ABA, auxin, cytokinin and ethylene) may play an important role in regulating the shift between anisohydry and isohydry. This provides a solid basis for future research. A better understanding of the molecular underpinnings of drought tolerance in wheat is important, as it will lead to a better understanding of the underlying physiology. This may lead to improved phenotyping strategies, ultimately leading to better selection of drought-tolerant wheat germplasm.

## Figures and Tables

**Figure 1 genes-12-01742-f001:**
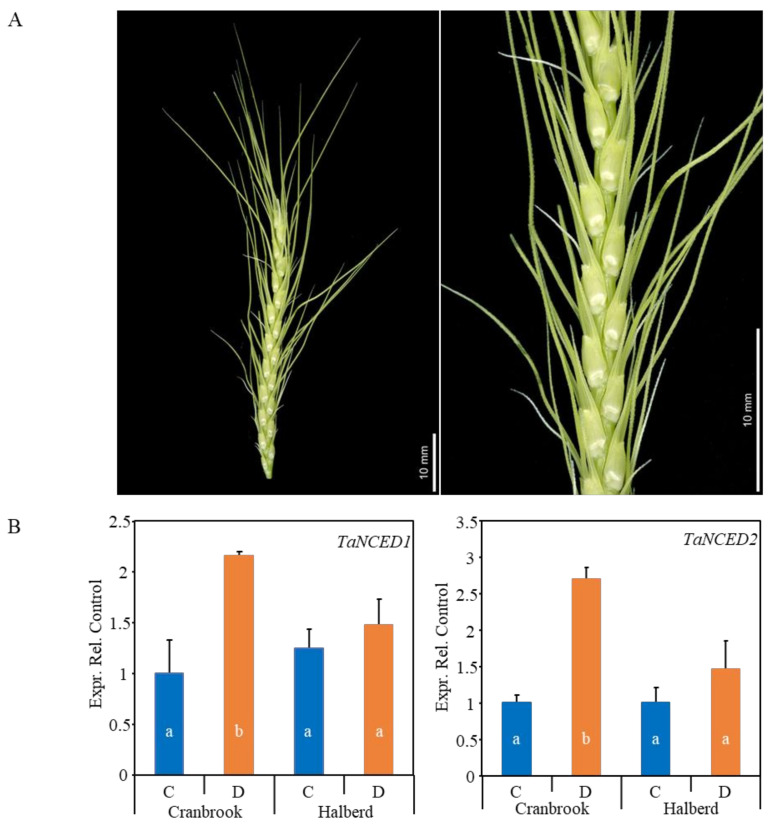
(**A**) The picture shows a YM stage wheat spike dissected from the leaf sheaths. The close-up and inset shows the spikelets and florets respectively at the time of pollen meiosis. Plants were drought-stressed at this stage and dissected spike material was used for transcriptome and hormone analysis. (**B**) Real-time PCR expression analysis of the wheat *TaNCED1* and *TaNCED2* ABA biosynthetic genes in unstressed control (C) and drought-stressed (D) Cranbrook and Halberd spikes. Each sample consisted of three repeats and error bars represent standard errors. Bars in the graph labelled with different letters are significantly different compared to the Cranbrook unstressed control (C) (*t*-test; *p* < 0.05).

**Figure 2 genes-12-01742-f002:**
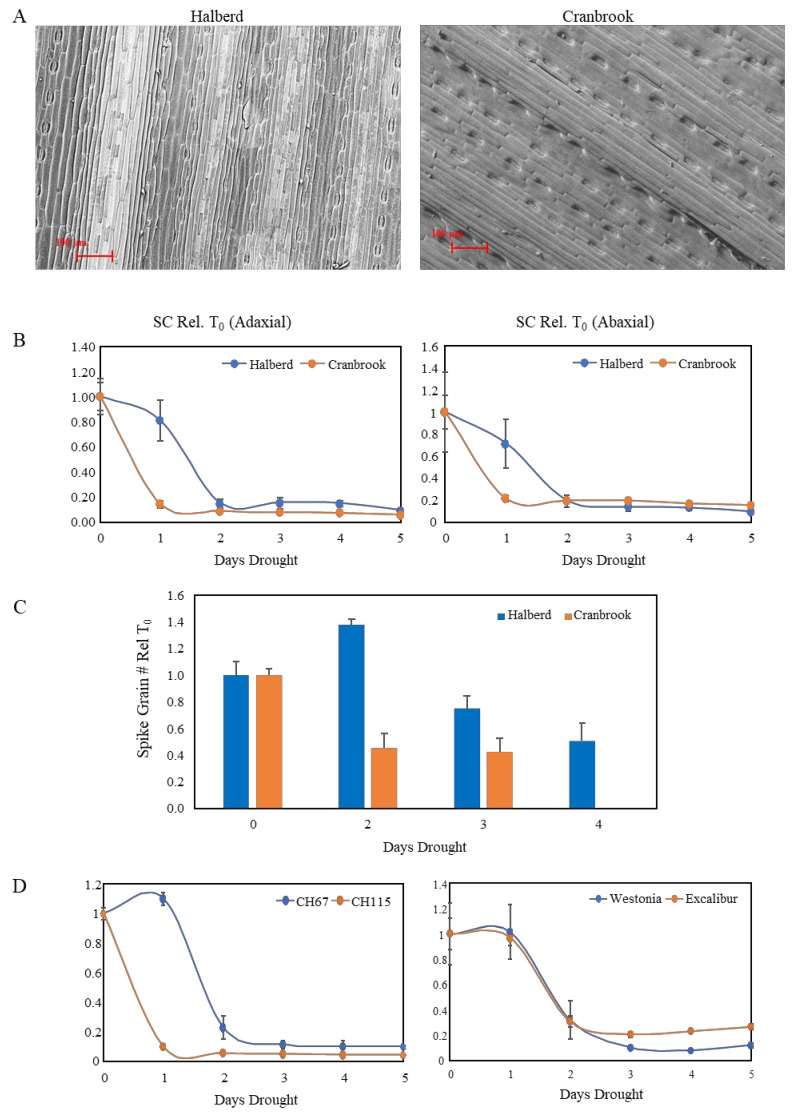
(**A**) Scanning electron microscope pictures of Cranbrook and Halberd flag leaves. The pictures were used to measure stomatal density and length of the guard cells using Fiji software. (**B**) Stomatal conductance (SC) measurements of Cranbrook and Halberd flag leaves over a period of 5-days of drought treatment. Five porometer readings were taken at each time point and results were averaged (error bars are standard errors). The data are expressed relative to the T_0_ unstressed control plants. (**C**) Effect of drought stress on Cranbrook and Halberd spike grain numbers. Plants from the 5-day drought time course experiment were re-watered at different time points and allowed to develop to maturity. Spike grain numbers were determined for 5-10 spikes per time point and spike grain numbers were expressed relative to unstressed T_0_ numbers. Error bars represent standard errors. (**D**) SC measurements for a 5-day drought stress time course experiment for a drought-tolerant (CH67) and a drought-sensitive (CH115) tail line of a Cranbrook × Halberd DH population (left) and two additional wheat varieties with known reproductive stage drought tolerance (Excalibur and Westonia; right).

**Figure 3 genes-12-01742-f003:**
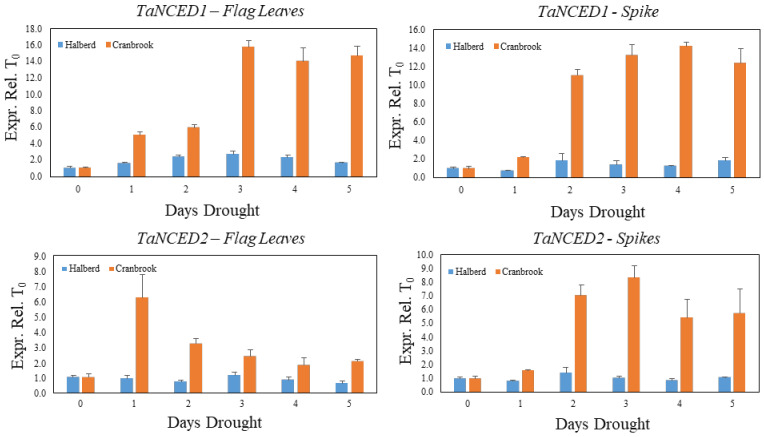
Evolution of *TaNCED1* and *TaNCED2* gene expression in Cranbrook and Halberd flag leaves and spikes during the 5-day drought time course experiment (Figure 2). Three repeat samples were tested per time point and error bars represent standard errors.

**Figure 4 genes-12-01742-f004:**
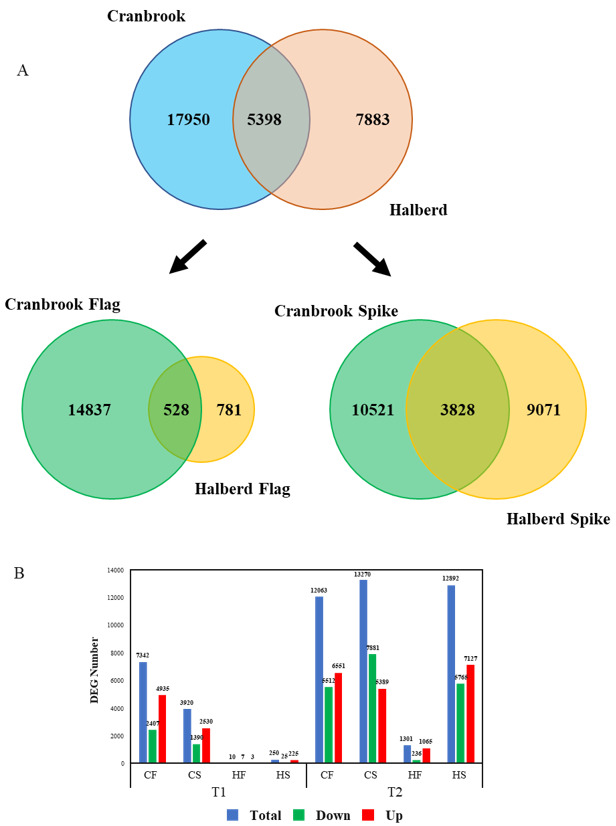
(**A**) Quantitative analysis of the transcriptome data using Venn diagrams. The diagrams represent the comparison of the total DEG numbers for Cranbrook and Halberd flag leaves and spikes (top) and the flag leaf and spike DEG numbers for each separately (bottom graph). (**B**) Break-up of the DEG numbers in total, up-regulated and down-regulated gene numbers at each time point for Cranbrook (C) and Halberd (H) flag leaves (HF, CF) and spikes (HS, CS).

**Figure 5 genes-12-01742-f005:**
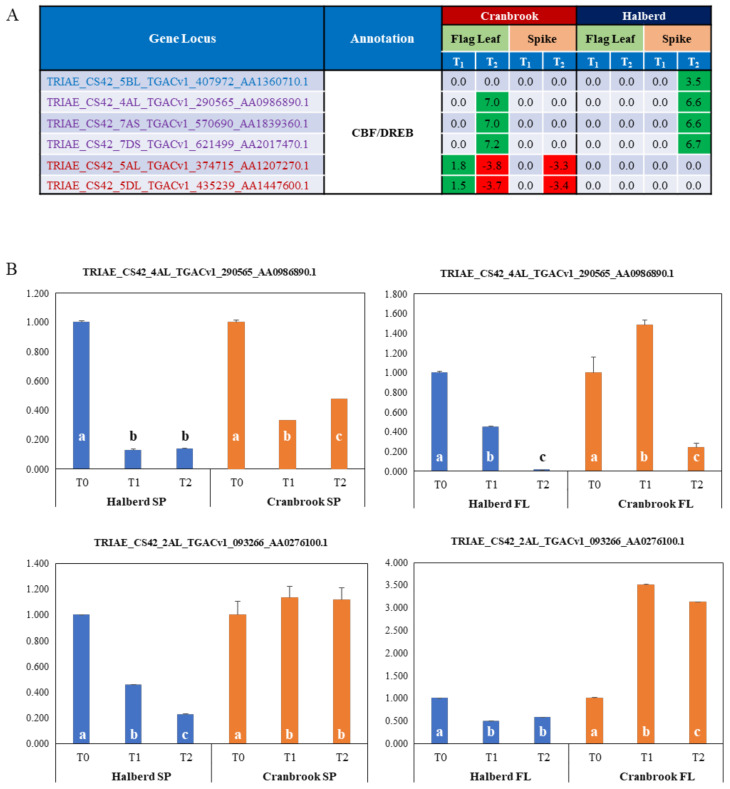
(**A**) List of potential CBF/DREB encoding genes that are significantly enriched in Cranbrook and Halberd flag leaves and spikes, and their LogFC expression values relative to unstressed plants (green = up-regulated; red = down-regulated). Gene loci in blue and red are specifically enriched in Halberd and Cranbrook respectively and loci in purple are enriched in both wheat lines. (**B**) Real-time PCR gene expression analysis of two CBF/DREB genes, including one enriched gene the list in A and another non-enriched CBF/DREB gene (see Appendix A). Data are the average of three repeat samples and error bars represent standard errors. Gene expression levels are relative to the unstressed control (=1) for each wheat line and bars in the graph labelled with different letters are significantly different (*t*-test; *p* < 0.05).

**Figure 6 genes-12-01742-f006:**
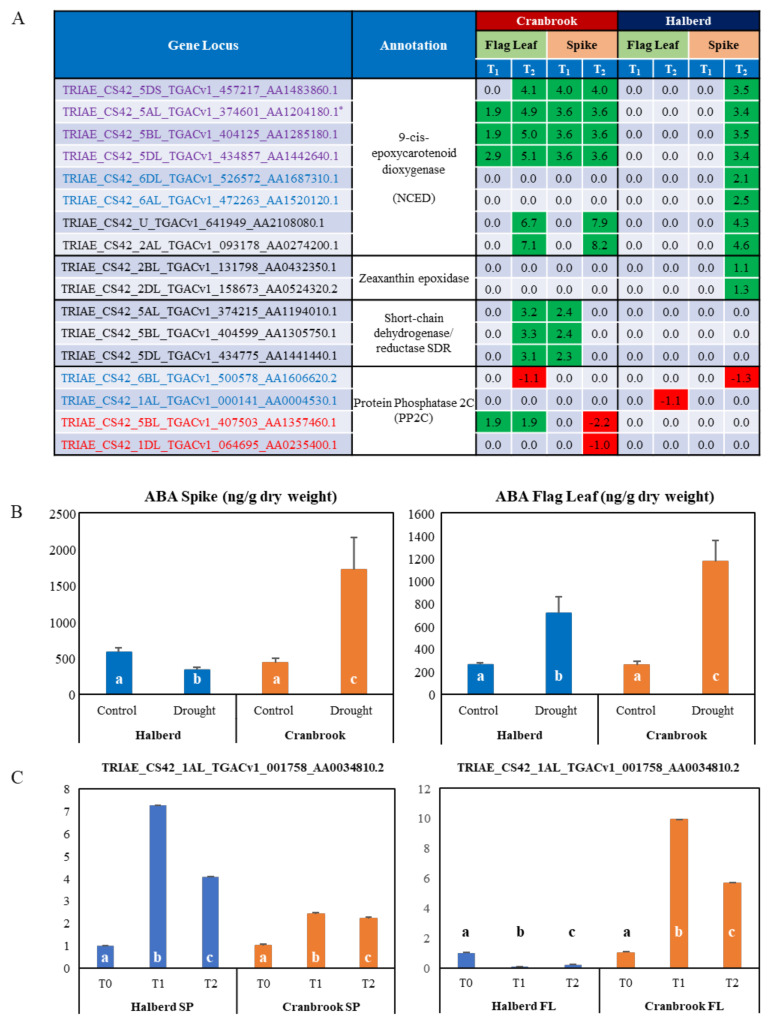
(**A**) List of genes involved in ABA synthesis and signaling. The NCED and PP2C genes were significantly enriched; gene loci in blue and red are specifically enriched in Halberd and Cranbrook, loci in purple are enriched in both wheat lines. Other potential ABA biosynthetic genes encoding zeaxanthin epoxidase and short-chain dehydrogenase/reductases were also amongst the DEG and were added to the list. The *TaNCED2* gene used for expression analysis in Figure 1B is marked with an asterisk. The expression data are listed as Log FC values relative to unstressed plants (green = up-regulated; red = down-regulated). (**B**) Determination of ABA concentrations in control and drought-stressed Halberd and Cranbrook flag leaves and spikes. Average ABA levels are expressed in ng/g dry weight (Y-axis). Three repeats were used for each measurement and error bars are standard errors. ABA levels that are significantly different compared the Halberd unstressed control and compared to the other samples in the graph are labelled with different letters (*t*-test; *p* < 0.05). (**C**) Real-time PCR expression studies of one candidate bZIP transcription factor gene. Three repeat samples were tested and error bars show standard errors. FL = flag leaf; SP = spike. Gene expression levels are relative to the unstressed control (=1) for each wheat line and bars in the graph labelled with different letters are significantly different (*t*-test; *p* < 0.05).

**Figure 7 genes-12-01742-f007:**
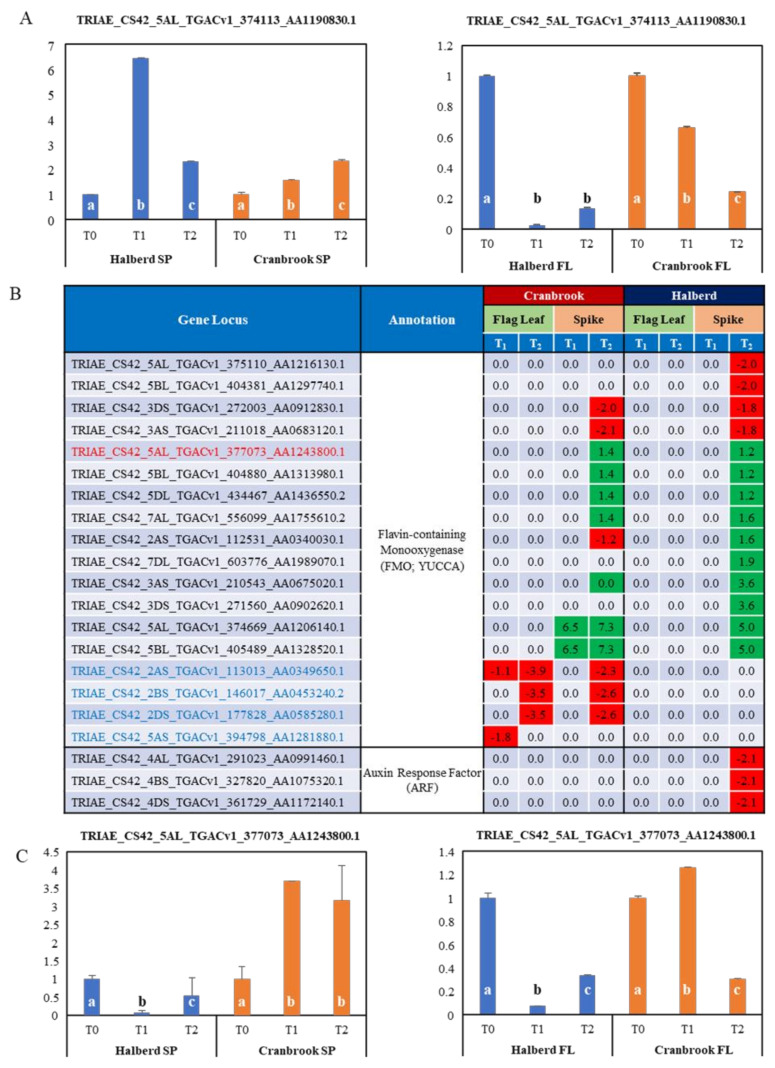
(**A**) Real-time PCR expression studies of one candidate Small Auxin Up-Regulated (SAUR) gene. Three repeat samples were tested and error bars show standard errors. Gene expression levels are relative to the unstressed control (=1) for each wheat line and bars in the graph labelled with different letters are significantly different (*t*-test; *p* < 0.05). (**B**) List of enriched genes potentially encoding flavin-containing monooxygenase (FMO; YUCCA; 18 genes) involved in auxin biosynthesis, and genes encoding Auxin Response Factors (ARF; three genes) transcription factors. The expression data are listed as Log FC values relative to unstressed plants (green = up-regulated; red = down-regulated). (**C**) Real-time PCR expression studies of one candidate *FMO* gene. Three repeat samples were tested and error bars show standard errors. Gene expression levels are relative to the unstressed control (=1) for each wheat line and bars in the graph labelled with different letters are significantly different (*t*-test; *p* < 0.05).

**Figure 8 genes-12-01742-f008:**
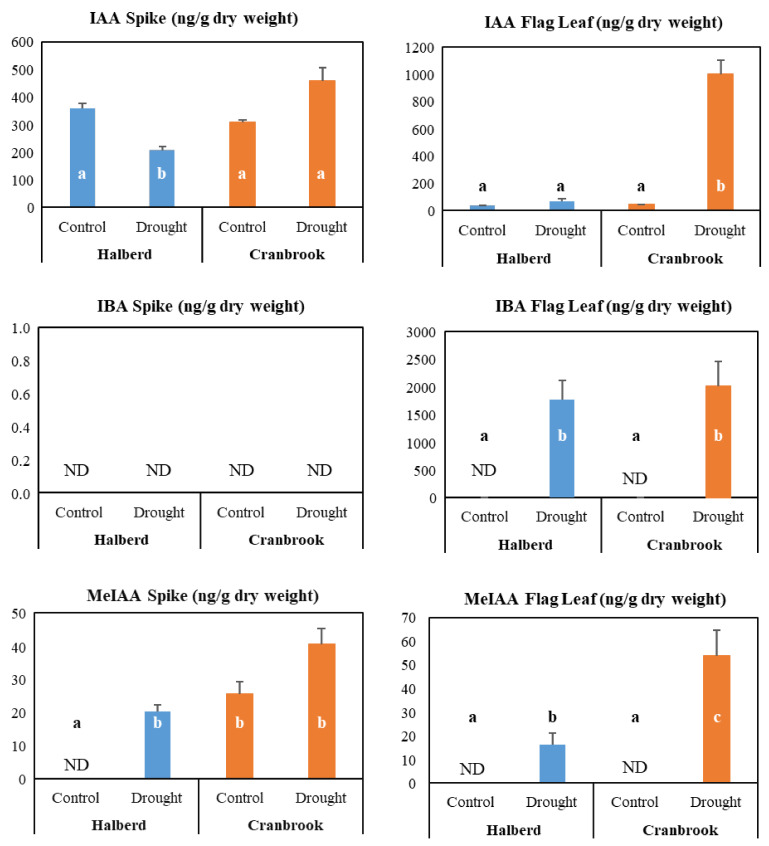
Measurements of indole-3-acetic acid (IAA), indole-3-butyric acid (IBA) and methyl-indole-3-acetic acid (MeIAA) in control and drought-stressed Halberd and Cranbrook flag leaves and spikes. Average ABA levels are expressed in ng/g dry weight (Y-axis). Three repeats were used for each measurement and error bars are standard errors. Hormone levels that are significantly different compared the Halberd unstressed control and and compared to the other samples in the graph are labelled with different letters (*t*-test; *p* < 0.05). For samples labelled “ND” hormone levels were not detectable.

**Figure 9 genes-12-01742-f009:**
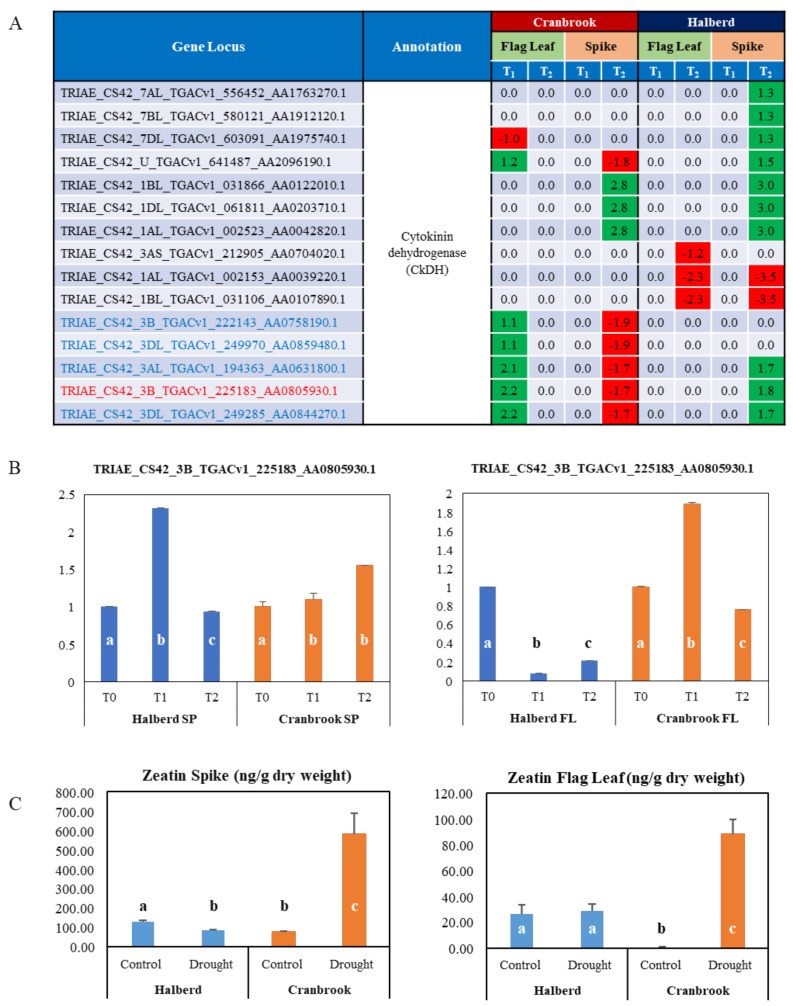
(**A**) Expression data for 15 enriched genes encoding cytokinin dehydrogenase (CkDH). Gene loci in blue and red are specifically enriched in Halberd and Cranbrook, loci in purple are enriched in both wheat lines. The expression data are listed as Log FC values relative to unstressed plants (green = up-regulated; red = down-regulated). (**B**) Expression analysis of one candidate *TaCkDH* gene. Three repeat samples were tested, and expression data are relative to unstressed control expression levels (=1). Error bars show standard errors. FL = flag leaf; SP = spike. Gene expression levels are relative to the unstressed control (=1) for each wheat line and bars in the graph labelled with different letters are significantly different (*t*-test; *p* < 0.05). (**C**) Measurement of zeatin levels in control and drought-stressed Halberd and Cranbrook flag leaves and spikes. Average ABA levels are expressed in ng/g dry weight (Y-axis). Three repeats were used for each measurement and error bars are standard errors. Zeatin levels that are significantly different compared the Halberd unstressed control and compared to the other samples in the graph are labelled with different letters (*t*-test; *p* < 0.05).

**Table 1 genes-12-01742-t001:** GO enrichment analysis for biological processes (BP) using Halberd and Cranbrook flag leaf DEG.

Halberd Flag Leaf BP GO Enrichment						Cranbrook Flag Leaf BP GO Enrichment					
GO.ID	Term	Annotated	Significant	Expected	Classic	GO.ID	Term	Annotated	Significant	Expected	Classic
GO:0050794	regulation of cellular process	383	46	24.36	5.70 × 10^−6^	GO:0019538	protein metabolic process	514	316	248.09	2.00 × 10^−11^
GO:0050789	regulation of biological process	400	46	25.44	1.90 × 10^−5^	GO:0010467	gene expression	413	273	199.34	2.10 × 10^−15^
GO:0006355	regulation of transcription, DNA-templat…	264	38	16.79	5.20 × 10^−7^	GO:0006796	phosphate-containing compound metabolic …	413	257	199.34	4.90 × 10^−10^
GO:1903506	regulation of nucleic acid-templated tra…	264	38	16.79	5.20 × 10^−7^	GO:0016070	RNA metabolic process	337	212	162.66	6.20 × 10^−9^
GO:2000112	regulation of cellular macromolecule bio…	272	38	17.3	1.10 × 10^−6^	GO:0006468	protein phosphorylation	298	182	143.84	1.80 × 10^−6^
GO:0009790	embryo development	15	7	0.95	1.60 × 10^−5^	GO:0006355	regulation of transcription, DNA-templat…	264	167	127.43	1.90 × 10^−7^
GO:0006032	chitin catabolic process	22	7	1.4	0.00028	GO:0043043	peptide biosynthetic process	100	76	48.27	7.70 × 10^−9^
GO:0006869	lipid transport	20	6	1.27	0.00111	GO:0006633	fatty acid biosynthetic process	85	49	41.03	0.0497
GO:0019684	photosynthesis, light reaction	5	5	0.32	9.90× 10^−7^	GO:0015672	monovalent inorganic cation transport	31	24	14.96	0.00084
GO:0006086	acetyl-CoA biosynthetic process from pyr…	3	3	0.19	0.00025	GO:0006559	L-phenylalanine catabolic process	32	24	15.45	0.00182
GO:0006094	gluconeogenesis	3	3	0.19	0.00025	GO:0072528	pyrimidine-containing compound biosynthe…	19	14	9.17	0.02206
GO:0006099	tricarboxylic acid cycle	3	3	0.19	0.00025	GO:0046856	phosphatidylinositol dephosphorylation	13	12	6.27	0.00113
GO:0006561	proline biosynthetic process	3	3	0.19	0.00025	GO:0016192	vesicle-mediated transport	17	12	8.21	0.05353
GO:0006816	calcium ion transport	4	3	0.25	0.00097	GO:0006563	L-serine metabolic process	15	11	7.24	0.04451
GO:0006415	translational termination	5	3	0.32	0.0023	GO:0006081	cellular aldehyde metabolic process	8	8	3.86	0.00291
GO:0009269	response to desiccation	6	3	0.38	0.00438	GO:0006544	glycine metabolic process	8	7	3.86	0.02803
GO:0006183	GTP biosynthetic process	8	3	0.51	0.01116	GO:0006656	phosphatidylcholine biosynthetic process	6	6	2.9	0.01257
GO:0006228	UTP biosynthetic process	8	3	0.51	0.01116	GO:0009107	lipoate biosynthetic process	6	6	2.9	0.01257
GO:0015696	ammonium transport	8	3	0.51	0.01116	GO:0009269	response to desiccation	6	6	2.9	0.01257
GO:0006536	glutamate metabolic process	11	3	0.7	0.02855	GO:0042128	nitrate assimilation	6	6	2.9	0.01257
GO:0006813	potassium ion transport	13	3	0.83	0.04507	GO:0006006	glucose metabolic process	5	5	2.41	0.0261
GO:0015743	malate transport	3	2	0.19	0.01156	GO:0006012	galactose metabolic process	5	5	2.41	0.0261
GO:0006730	one-carbon metabolic process	6	2	0.38	0.05091	GO:0007205	protein kinase C-activating G protein-co…	5	5	2.41	0.0261
						GO:0009101	glycoprotein biosynthetic process	5	5	2.41	0.0261
						GO:0006353	DNA-templated transcription, termination	4	4	1.93	0.05415
						GO:0006680	glucosylceramide catabolic process	4	4	1.93	0.05415
						GO:0015914	phospholipid transport	4	4	1.93	0.05415
						GO:0032957	inositol trisphosphate metabolic process	4	4	1.93	0.05415
						GO:0042372	phylloquinone biosynthetic process	4	4	1.93	0.05415
						GO:0046168	glycerol-3-phosphate catabolic process	4	4	1.93	0.05415

GO terms highlighted in red were referred to in the Results section. The ‘Annotated’, ‘Significant’ and ‘Expected’ columns refer to the total number of genes covered by each GO term, the number of those genes that were significantly enriched by cold treatment, and the expected frequency of these genes under normal conditions, respectively. The ‘classic’ column lists the Fisher’s exact test results for each GO term. All Go terms with *p* < 0.05 were considered significant and listed in ascending order in this table.

**Table 2 genes-12-01742-t002:** GO enrichment analysis for biological processes (BP) using all Halberd and Cranbrook spike DEG.

Halberd Spike BP GO Enrichment						Cranbrook Spike BP GO Enrichment					
GO.ID	Term	Annotated	Significant	Expected	Classic	GO.ID	Term	Annotated	Significant	Expected	Classic
GO:0050896	response to stimulus	372	221	167.73	1.90 × 10^−9^	GO:0051179	localization	451	260	230.41	0.00137
GO:0005975	carbohydrate metabolic process	185	107	83.41	0.00021	GO:0051234	establishment of localization	449	258	229.39	0.00186
GO:0008610	lipid biosynthetic process	99	59	44.64	0.00222	GO:0006810	transport	449	258	229.39	0.00186
GO:0007017	microtubule-based process	60	49	27.05	4.50 × 10^−9^	GO:0015979	photosynthesis	55	52	28.1	1.50 × 10^−12^
GO:0007018	microtubule-based movement	59	48	26.6	8.30 × 10^−9^	GO:0042592	homeostatic process	39	31	19.92	0.00022
GO:0009057	macromolecule catabolic process	65	47	29.31	6.40 × 10^−6^	GO:0006820	anion transport	40	28	20.44	0.01146
GO:0009733	response to auxin	58	41	26.15	6.10 × 10^−5^	GO:0045454	cell redox homeostasis	34	26	17.37	0.00208
GO:0006073	cellular glucan metabolic process	51	35	22.99	0.00053	GO:0033014	tetrapyrrole biosynthetic process	23	23	11.75	1.80 × 10^−7^
GO:0007275	multicellular organism development	45	33	20.29	0.0001	GO:0006032	chitin catabolic process	22	19	11.24	0.00058
GO:0042546	cell wall biogenesis	46	27	20.74	0.04294	GO:0006418	tRNA aminoacylation for protein translat…	26	19	13.28	0.01862
GO:0034655	nucleobase-containing compound catabol…	25	18	11.27	0.00579	GO:0046148	pigment biosynthetic process	23	18	11.75	0.00692
GO:0006032	chitin catabolic process	22	17	9.92	0.00213	GO:0006814	sodium ion transport	16	16	8.17	2.10 × 10^−5^
GO:0006260	DNA replication	23	17	10.37	0.00474	GO:0006364	rRNA processing	15	14	7.66	0.00063
GO:0000160	phosphorelay signal transduction system	19	13	8.57	0.03435	GO:0046856	phosphatidylinositol dephosphorylation	13	12	6.64	0.00213
GO:0007049	cell cycle	12	12	5.41	6.90 × 10^−5^	GO:0016226	iron-sulfur cluster assembly	12	11	6.13	0.00389
GO:0009690	cytokinin metabolic process	15	12	6.76	0.00633	GO:0009082	branched-chain amino acid biosynthetic p…	10	10	5.11	0.00119
GO:0006308	DNA catabolic process	16	12	7.21	0.01497	GO:0006096	glycolytic process	7	7	3.58	0.00902
GO:0033875	ribonucleoside bisphosphate metabolic pr…	11	11	4.96	0.00015	GO:1901663	quinone biosynthetic process	6	6	3.07	0.01769
GO:0034032	purine nucleoside bisphosphate metabolic…	11	11	4.96	0.00015	GO:0006006	glucose metabolic process	5	5	2.55	0.03469
GO:0009269	response to desiccation	6	6	2.71	0.00835	GO:0006817	phosphate ion transport	5	5	2.55	0.03469
GO:0015969	guanosine tetraphosphate metabolic proce…	6	6	2.71	0.00835	GO:0007205	protein kinase C-activating G protein-co…	5	5	2.55	0.03469
GO:0006002	fructose 6-phosphate metabolic process	5	5	2.25	0.01855	GO:0046836	glycolipid transport	5	5	2.55	0.03469
GO:0006275	regulation of DNA replication	5	5	2.25	0.01855						
GO:0007205	protein kinase C-activating G protein-co…	5	5	2.25	0.01855						
GO:0006635	fatty acid beta-oxidation	4	4	1.8	0.04122						

GO terms highlighted in red were referred to in the Results section. The ‘Annotated’, ‘Significant’ and ‘Expected’ columns refer to the total number of genes covered by each GO term, the number of those genes that were significantly enriched by cold treatment, and the expected frequency of these genes under normal conditions, respectively. The ‘Classic’ column lists the Fisher’s exact test results for each GO term. All Go terms with *p* < 0.05 were considered significant and listed in ascending order in this table.

## Data Availability

All experimental data are made available as Appendix A files attached to this manuscript.

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
