# Peer review of "Reproductive Stage Drought Tolerance in Wheat: Importance of Stomatal Conductance and Plant Growth Regulators"

_genes, 2021, doi:10.3390/genes12111742_

Round 1
Reviewer 1 Report
This is an interesting paper that offers a useful and comprehensive insight into certain physiological and molecular mechanisms underlying the drought response strategies in wheat during the reproductive stage. The research has been well done and has obtained many valuable and meaningful results. Generally, the paper is well written, the presentation in the Results is clear, and the Discussion gives a consistent interpretation of the main results.
The eventual weakness of the manuscript is the involvement of the additional wheat lines only in the SC measurements, which may raise the question of why these lines are not present in all analyses. My impression is that this addition doesn't add much to the quality of the paper and that keeping a straight line in the experiments with the drought-tolerant Halberd and drought-sensitive Cranbrook could be better (and that comparison of SC with other lines can be eventually used for Discussion). Although the authors can consider this suggestion, it is not obligatory.
Other issues that need the author's attention:
Line 55: what "genes measured up" refers to - field performance of transgenic lines or something else? Please be more precise.
Line 71: reference number 34 is doubled.
Line 128: sentence "Measurements were carried out for both the abaxial and adaxial stomata." is unnecessary, since from the next one is clear that stomatal conductance was measured on both leaf's sides.
Lines 136-138: Please avoid repetition that samples were collected from both control and drought-stressed plants in these two sentences.
Line 141: RNA should be precipitated using LiCl, but this is not clearly stated.
Line 187: in section 2.4. Phytohormone quantification, please name each of the plant growth regulators that are analyzed.
Lines 216-217: the last sentence is more appropriate for Discussion.
Line 240: "grain loss was reduced by ~100% (Fig. 2C)" or grain loss was ~100% (Fig. 2C)?
Lines 503-508: these statements do not belong to the Results and are more appropriate for Discussion.
Line 665: Please indicate what "WUE" means.
Line 689: "CE" stands for?
Figure 5B, Figure 6B and C, Figure 7A and C, Figure 8 and Figure 9B and C: distribution of letters that indicate statistically significant differences are confusing. Here, among statistically different values "a" should be used for the lowest, while "c" for the highest one (or opposite). Also, for instance in Figure 7C (TRIAE_CS42_5AL_TGACv1_377073_AA1243800.1) the same letter is used for the lowest (Halberd FL - T1) and highest (Cranbrook FL - T1) value, while values between them are statistically different. Or in Figure 8 more than a 5-fold increase in Flag Leaf IAA for Cranbrook during the drought appears to be insignificant. Additionally, the y-axis needs a title and the parts of some graphs (TRIAE_CS42_4AL_TGACv1_290565_AA0986890.1, TRIAE_CS42_1AL_TGACv1_001758_AA0034810.2, MeIAA Spike, TRIAE_CS42_3B_TGACv1_225183_AA0805930.1) are grey instead of black. Finally, IBA Spike in Fig.8 looks "empty". Please indicate that IBA is not detected (for instance with "ND" in the places for columns, with the explanation of this abbreviation in the figure legend).
Author Response
Thank you for reviewing our paper. The response to the comments raised are attached in the uploaded word document.

Reviewer 2 Report
The current manuscript is interesting and the experiments are well designed and the analysis of two different wheat lines who show opposite drought phenotypes is indeed intriguing
I do however have few comments for the authors:
1) The figures are well described but they are all missing the legends making difficult to follow the data presented. Please include them.
2)Figure 1A: the authors mentioned that chlorophyll content is low however they are not presenting any quantitative data. I would suggest to include this analysis.
3)Figure 4A: The authors show a Venn diagram describing the number of the total DEGs between Cranbrook and Halberg. I think it is more informative to describe the DEGs per time point rather than the total number.
4) The results are all very interesting but I rather feel that some of them could be moved to the supplement and focusing the manuscript on one specific pathway.
Author Response
Thank you for taking time to review our research article. The response to the comments is addressed in the document attached.
Kind regards,
Olive
